# ReForm-Eval: Evaluating Large Vision Language Models via Unified Re-Formulation of Task-Oriented Benchmarks

## ABSTRACT

Recent years have witnessed remarkable progress in the development of large vision-language models (LVLMs). Benefiting from the strong language backbones and efficient cross-modal alignment strategies, LVLMs exhibit surprising capabilities to perceive visual signals and perform visually grounded reasoning. However, the capabilities of LVLMs have not been comprehensively and quantitatively evaluated. Most existing multi-modal benchmarks require task-oriented input-output formats, posing great challenges to automatically assess the free-form text output of LVLMs. To effectively leverage the annotations available and reduce the manual efforts required for constructing new benchmarks, we propose to re-formulate existing benchmarks into unified LVLM-compatible formats. Through systematic data collection and reformulation, we present ReForm-Eval benchmark, offering substantial data for evaluating various capabilities of LVLMs. Through extensive experiments and analysis in ReForm-Eval, we demonstrate the comprehensiveness and reliability of ReForm-Eval in assessing various LVLMs. Our benchmark and evaluation framework will be open-sourced as a cornerstone for advancing the development of LVLMs.

## CCS CONCEPTS

• **Computing methodologies → Computer vision tasks**; **Natural language generation**.

## KEYWORDS

large vision language model, multi-modal benchmark, evaluation

## 1 INTRODUCTION

With the trend led by ChatGPT [62], LLMs (Large Language Models) [13, 63, 78] have ushered in revolutionary advancements of Natural Language Processing (NLP). Inspired by these efforts, many researchers attempt to extend the success of LLMs to the realm of vision and language. By equipping LLMs with visual encoders and aligning multi-modal representations through generative pre-training, large vision-language models (LVLMs) [5, 10, 41, 46, 50, 52, 55, 76, 91, 101] possess the capability to comprehend visual information and engage in multi-modal conversations with users.

Despite the potential shown by LVLMs to become the general-purpose foundation models for multimedia information processing, the reliability of LVLMs in various scenarios still hangs in doubt. On the one hand, LVLMs demonstrate surprising abilities

**Unpublished working draft. Not for distribution.**

Permission to make digital or hard copies of all or part of this work for personal or classroom use is granted without fee provided that copies are not made or distributed for profit or commercial advantage and that copies bear this notice and the full citation on the first page. Copyrights for components of this work owned by others than the author(s) must be honored. Abstracting with credit is permitted. To copy otherwise, or republish, to post on servers or to redistribute to lists, requires prior specific permission and/or a fee. Request permissions from permissions@acm.org.
*ACM MM, 2024, Melbourne, Australia*
© 2024 Copyright held by the owner/author(s). Publication rights licensed to ACM.
ACM ISBN 978-x-xxxx-xxxx-x/YY/MM
https://doi.org/10.1145/nnnnnnn.nnnnnnn

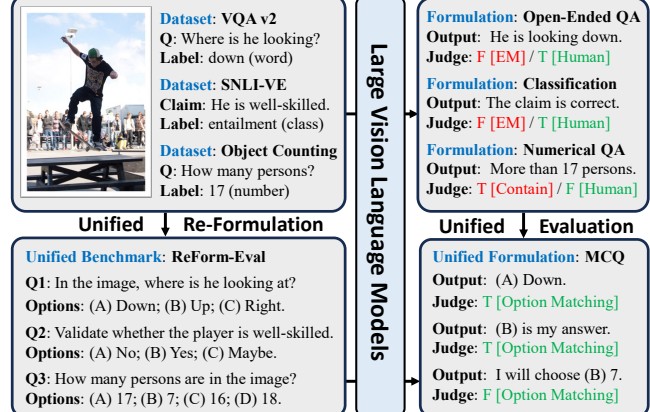

**Figure 1: Illustration of the unified framework of re-formulating existing benchmarks into multiple-choice questions (MCQ). The text within square brackets indicates the evaluation methods, with red and green denoting incorrect and correct judgment, respectively. Q, T, F, EM stand for Question, True, False, and Exact Match, respectively.**

like OCR [26, 55], meme understanding [1, 101], and visual commonsense reasoning [41, 51]. On the other hand, LVLMs suffer from serious hallucination issues [45, 49, 84]. To comprehensively evaluate LVLMs, endeavors have been made to construct new benchmarks [18, 40, 53, 95, 96]. However, the additional costs associated with manual data collection and annotation limit these benchmarks in terms of quantity and scope, making further extension challenging. Meanwhile, there exist affluent task-oriented datasets covering various scenarios, but these datasets cannot be directly applied to assess LVLMs, leading to a waste of data resources.

The main reason behind this situation is the structural gap between existing task-oriented multi-modal benchmarks and LVLMs. Most existing benchmarks are designed for specific tasks and demand highly structured input-output formats [29, 34, 47, 86]. For instance, VQA v2 [22] requires concise answers, typically in the form of single words or short phrases. Previously evaluated vision-language pre-trained models [12, 99] need to be fine-tuned and learn task-specific parameters to fit the structures of such benchmarks. On the contrary, LVLMs are flexible and tend to provide detailed responses [50]. As depicted in the upper flowchart of Figure 1, such a structural gap makes the automated evaluation criteria unstable and varied. For example, regarding the model's response to the VQA v2 example, "he is looking down" is rejected by the EM method but actually accepted by humans. This poses the greatest obstacle to accurate automated evaluation, particularly when assessing the desired zero-shot capabilities of LVLMs.

In this paper, we aim to fully utilize existing resources to evaluate LVLMs. To bridge the structural gap, we explore ways of **re-formulating existing benchmarks into unified formats that**

| Benchmark | Size | Annotation | | Evaluation | | Instability | | | Instability Measure |
|---|---|---|---|---|---|---|---|---|---|
| | | Human | ChatGPT | ChatGPT | Unified Form | Instruction | Option Mark | Option Order | |
| **LAMM** [92] | 186,000 | | ✓ | ✓ | | | | | None |
| **MME** [18] | 2,374 | ✓ | | | ✓ | | | | None |
| **LVLM-eHub** [88] | 1,242,830 | | | ✓ | | ✓ | | | None |
| **MMBench** [53] | 2,974 | ✓ | | ✓ | ✓ | | | ✓ | $\Delta acc$ |
| **MMMU** [96] | 11,550 | ✓ | | | ✓ | | | | None |
| **MMVet** [95] | 218 | ✓ | | ✓ | ✓ | | | | None |
| **ReForm-Eval** | 521,712 | | ✓ | | ✓ | ✓ | ✓ | ✓ | entropy |

Table 1: Comparison with existing evaluation benchmarks. The term "unified form" denotes a standardized evaluation format. In MMBench [53], the option order instability is measured by the difference between the accuracy $\Delta acc$ from CircularEval and VanillaEval. While in Reform-Eval, we measure the instability by the entropy of the prediction distribution (see Section 4.2).

**are compatible with LVLMs**. Referring to Figure 1, we adapt the data and evaluation process to the unified form shown in the lower part. Firstly, we propose an automatic framework to re-formulate existing datasets into either multiple-choice questions or text generation problems. These two forms of re-formulation conform to the flexible textual output of LVLMs [25]. For each dataset, the choice of re-formulation format is determined by considering its corresponding tasks. For tasks with specific text generation requirements, like OCR and image captioning, datasets are re-formulated as specialized text generation problems, while other datasets are restructured into multiple-choice problems.

The unified formulation further enables consistent evaluation. We design a reliable evaluation method that considers both the input sensitivity and the output control of LVLMs, alleviating the requirement for assistance from ChatGPT or human. As mentioned in [18], current LVLMs struggle to follow multiple-choice instructions. We propose two approaches to mitigate this issue: (1) **Black-box** approach: Guiding LVLMs to output in desired formats through in-context-learning; (2) **White-box** approach: Directly calculating the generation probability for options and selecting the one with the highest value. With regard to the sensitivity of LVLMs to the input prompts [98], we design **an instability-aware evaluation strategy** and introduce a metric to characterize such instability.

Based on the re-formulation framework, we present our unified multi-modal benchmark, ReForm-Eval. For a comprehensive evaluation, we re-formulate 61 benchmark datasets based on existing data resources, the evaluation dimensions range from basic visual perception to high-level visual reasoning and dialog. Compared with recent LVLM benchmarks that require manual annotation [18, 53, 95], ReForm-Eval fully utilizes publicly open resources and provides significantly more data, almost 100 times the size of MMBench [53]. Meanwhile, unlike LVLM-ehub [88], which requires designing complex and dataset-specific evaluation strategies, ReForm-Eval offers greater scalability. Generally speaking, ReForm-Eval is large-scale, easy to use, and provides a universally applicable and efficient evaluation approach, as shown in Table 34.

Based on ReForm-Eval, we conduct a comprehensive evaluation of existing LVLMs. Experiments demonstrate that ReForm-Eval and the proposed evaluation methods provide reliable evaluation results for a wide range of models. We hope ReForm-Eval constitutes a valuable augmentation to the ongoing efforts in LVLM research and could facilitate better development of LVLMs.

## 2 RELATED WORKS

### 2.1 Large Vision Language Models

Inspired by the advancements of LLMs and the multi-modal understanding abilities demonstrated by GPT-4 [63], developing open-source LVLMs currently dominates the multi-modal research. Visual signals encoded by visual encoders [66] are incorporated in LLMs through linear projection [80], Q-former [41], or cross-attention layers [4]. Most LVLMs are trained in two phases, pre-training and instruct tuning. Pre-training data involves image-text pairs [47, 68, 70] and multi-modal interleaved documents [102], multi-modal representations are aligned by training LVLMs to generate texts based on visual contents. To enable multi-modal instruct tuning, MiniGPT4 [101] bootstraps high-quality data by refining the previous output, LLaVA [52] proposes to employ GPT-4 to generate image-involved dialogs while other works construct instruct tuning data from existing vision-language benchmarks [15, 44, 89].

To seamlessly adapt LLMs for multi-modal scenarios, many efforts are paid including designing strategies for parameter freezing [91], introducing light-weight trainable modules into the backbone [19, 21], incorporating continuous output [9, 65], and enhancing the visual representations [26, 43, 46, 51, 98]. Benefiting from the aligned representations from ImageBind [20], LVLMs can be further extended to more modalities [24, 75].

### 2.2 Multi-Modal Benchmarks

*Task-Oriented Benchmarks.* Most existing benchmarks are designed for specific multi-modal tasks. They can not be directly utilized to evaluate LVLMs since they rely on structured input-output formats for evaluation. VQA v2 [22] requires concise answers, retrieval benchmarks [47, 93] demand dense scores for all image-text pairs, VCR [97] provides coordinates to refer visual object in the question, and bounding box output is necessary for RefCOCO [34]. This characteristic renders the application of such benchmarks on evaluating the free-form text outputs of LVLMs, unless task-specific post-processing and evaluation methods are implemented [88, 92].

*Benchmarks for LVLMs.* To facilitate reliable and efficient automated evaluation of LVLMs, efforts are paid to construct LVLM-compatible benchmarks, such as yes-or-no problems in MME [18] and multiple-choice problems in MMBench [40, 53]. A portion of the benchmarks are designed to assess specific capabilities [54, 83]

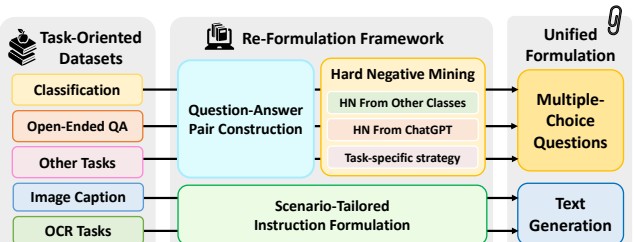

**Figure 2: The construction pipeline of ReForm-Eval.**

or diagnose particular issues [45, 100], while others aim for comprehensive evaluation [18, 53, 95, 96]. However, limited manual annotation (around 100 samples per dimension in MME and MM-Bench) could potentially introduce evaluation bias into the results.

## 3  REFORM-EVAL BENCHMARK

In this section, we describe the construction pipeline of ReForm-Eval (as shown in Figure 2). We start by introducing the general framework of our re-formulation process in Section 3.1. Then, Section 3.2 summarizes the capability dimensions assessed in ReForm-Eval and the corresponding datasets. The basic statistics of ReForm-Eval are detailed in Section 3.3.

### 3.1  Unified Re-Formulation Framework

Existing LVLMs primarily adopt LLMs as backbones and use free-form text to interact with users. This paradigm makes the output more flexible and aligned with human needs. However, the gap between these models and existing highly structured benchmarks poses challenges for evaluation. In order to effectively reuse the annotations in existing benchmarks, these benchmarks need to be re-formulated into appropriate formats. Motivated by benchmarks for LLMs [25, 27, 74], ReForm-Eval considers two formats that are compatible with LVLMs, namely multiple-choice problems and text generation problems.

Multiple-choice problem is the primary format in ReForm-Eval. By providing options for the questions, models are guided to produce responses in a constrained format. The key in multiple-choice problem construction is how to prepare meaningful negative options. Generally, for close-vocabulary classification tasks, we build relationships between categories based on which hard negative options are selected. For open-ended tasks, based on the question and the correct answer, negative options can be obtained with the help of task-specific strategies or LLMs like ChatGPT [62].

For OCR and image captioning that involves text generation, corresponding benchmarks are formulated as text generation problems tailored to various scenarios. We curate the input prompts to describe the tasks and requirements. For OCR tasks, responses should contain the target tokens in the image. For description tasks, models should provide concise depictions of the visual content.

### 3.2  Capability Dimensions

To address the wide range of questions posed by users, LVLMs need to possess diverse capabilities. For a comprehensive evaluation, we curate 61 benchmark datasets from existing resources, summarizing the assessed capabilities into 2 major categories and

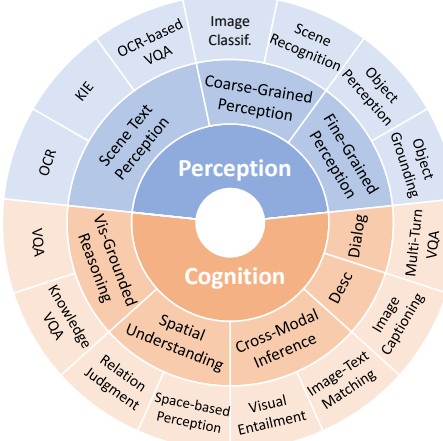

**Figure 3: Assessed capability dimensions and tasks in ReForm-Eval. "Desc" and "Classif" are short for visual description and classification, respectively.**

8 sub-categories which are illustrated in Figure 3. To avoid information overload, details about the re-formulation procedures and statistics of individual datasets are provided in Appendix A.

#### 3.2.1  Visual Perception Tasks.

**Coarse-Grained Perception (CG).** Coarse-grained perception is the ability to recognize the overall layout and main objects at the image level. We evaluate this capability through **image classification** (IC) using Flowers102 [61], CIFAR10 [37], ImageNet-1K [17], Pets37 [64], and MEDIC [3] benchmarks, and **scene recognition** (SR) using TDIUC [32] and VizWiz [23] benchmarks. The samples are re-formulated as multiple-choice questions.

**Fine-Grained Perception (FG).** Fine-grained perception requires detailed sensing at the object level. We set up the **object perception** (OP) task (using TDIUC [32] and MSCOCO [47] benchmarks) and the **object grounding** (OG) task (using MSCOCO [47] and RefCOCO [94] benchmarks) for evaluation. Object perception measures how well an LVLM can identify local semantics, while object grounding assesses the ability to localize fine-grained objects. All tasks are formulated as multiple-choice questions.

**Scene Text Perception (STP).** Scene text perception enables models to identify, understand, and perform inference based on text in images. This evaluation is conducted through **optical character recognition** (OCR) using 6 benchmarks (including CUTE80 [67], IC15 [33], IIIT5K [59], COCO-Text [59], WordArt [87] and TextOCR [73]), **key information extraction** (KIE) using 3 benchmarks (including SROIE [28], POIE [38] and FUNSD [30]) and **OCR-based VQA** using 3 benchmarks (including TextVQA [72], DocVQA [58] and OCR-VQA [60]). We consider STP as a specialized text generation problem that requires the output from LVLMs to perfectly match the text in the image.

#### 3.2.2  Visual Cognition Tasks.

**Visually Grounded Reasoning (VGR).** A reliable LVLM is supposed to perform reasoning based on multi-modal contextual information. In order to assess such capability, we adopt the commonly applied **visual question answering** (VQA) task and its variant,

| Statistics | Number |
|---|---|
| Total Questions | 521712 |
| Total Dimensions / Tasks / Datasets | 8/15/61 |
| Multiple-Choice Questions | 415283 (79.6%) |
| Text Generation Questions | 106429 (20.4%) |
| Average question length / # words | 171.42 / 35.98 |
| Average # words in | |
| - references of Description tasks | 12.11 |
| - references in OCR-related tasks | 2.78 |
| Average option length / # words | 10.79 / 2.0 |
| Avg. / Max. / Min. # option | 3.8 / 2 / 7 |
| Total Images | 333388 |
| - Average image width | 616.27 |
| - Average image height | 554.23 |
| - Average image ratio | 1.26 |

**Table 2: Key statistics in ReForm-Eval.**

**knowledge-based visual question answer** (K-VQA), which further requires models to utilize internally stored knowledge. For vanilla VQA, we adopt VQA v2 [22], GQA [29], and Whoops [6]. As for KVQA, we consider 6 benchamrks including OK-VQA [57], ScienceQA [56], VizWiz [23], ViQuAE [39], A-OKVQA [69] and ImageNetVC [85]. The aforementioned benchmarks are re-formulated into multiple-choice questions.

**Spatial Understanding (Spatial).** Spatial understanding is the key to the real-life application of LVLMs on robots. This task requires a comprehensive understanding of both the object-object and object-observer relationship so as to make reasonable behaviors. We access such capability through **spatial relation judgment** (SRJ) using VSR [48] and MP3D-Spatial, a benchmark designed for embodied tasks in real-world environments, constructed from Matterport3D [7]. Additionally, we employ **Space-Based Reasoning** (SBR) through the CLEVR [31] benchmark. The SRJ task aims to accurately identify spatial relationships, forming a concept of where the ego is in space. The SBP task entails complex reasoning ability based on the understanding of spatial relationships. All samples are re-formulated as multiple-choice questions.

**Cross-Modal Inference (CMI).** A thorough comprehension of both visual and textual modalities is required to perform cross-modal inference on the relationship between images and texts. We consider two tasks, **image-text matching** (ITM) and **visual entailment** (VE). ITM requires models to measure the cross-modal similarities, including MSCOCO [47], WikiHow [35] and Winoground [77]. VE demands models to check whether the information is entailed across modalities, using SNLI-VE [86] and MOCHEG [90]. Both tasks are re-formulated as multiple-choice questions.

**Visual Description (Desc).** Visual description is an inherent capability of LVLMs as generative models. We adopt the **image captioning** task on MSCOCO [47], TextCaps [71], NoCaps [2], and Flickr30K [93] for evaluation. These datasets are formulated as text generation problems with the requirement of concise outputs.

**Multi-Turn Dialogue (Dialog).** Existing benchmarks primarily focus on single-turn conversation. ReForm-Eval evaluates the performance of LVLMs in multi-turn dialogues. We consider the

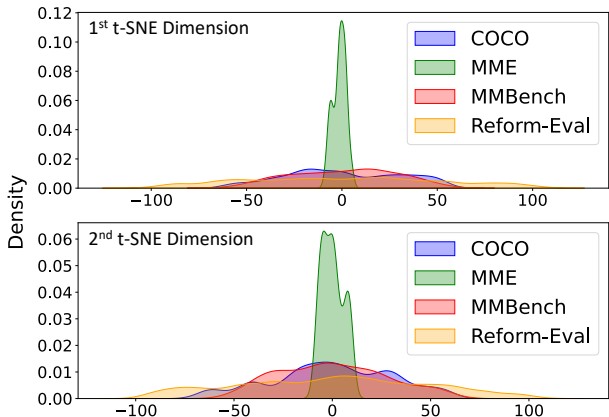

**Figure 4: Distribution of image embeddings encoded by CLIP-ViT-B/32 [66] from various benchmarks.**

**multi-turn VQA** task using VisDial [16] and VQA-MT, the latter is constructed by reorganizing questions in VQA v2. Both benchmarks are formulated as multiple-choice questions.

### 3.3 Dataset Statistics

The basic statistics of Reform-Eval are shown in Table 2. In general, there are 521,712 questions in our Reform-Eval benchmark with 79.6% as multiple-choice questions and 20.4% as text generation questions. The average lengths of questions and options are 36 words and 2 words, respectively. The average number of options is close to 4, with a minimum number of 2 for yes-or-no questions and a maximum number of 7 for disaster classification. To perform quality control, we manually sample and confirm the false negative rate of negative options generated by ChatGPT is below 0.01, while under other reformulation methods, it is 0. Please refer to Section 6.1 for the analysis of distractor construction methods.

The total number of images is ~333K, with an average ratio of 1.26. Figure 4 displays the distribution of images visualized with t-SNE [81]. Compared to previous benchmarks, the image distribution of ReForm-Eval is more extensive, allowing for a more comprehensive evaluation of LVLM across diverse visual scenarios.

## 4 EVALUATION STRATEGY

### 4.1 Evaluation Methods and Metrics

With the unified problem formulation, the performance of LVLMs can be universally assessed. For specialized generation problems, the evaluation method depends on the scenario. For visual description, we follow [41] to use CIDEr [82] as the main metric (more metrics are discussed in Appendix C). Since datasets mainly provide concise references, we craft prompts to require concise responses and limit the maximum number of tokens a model can generate. As for STP, input prompts are well-designed to instruct models to identify the scene texts. The metric is word-level accuracy: the proportion of ground-truth words that appear complete in the output.

Considering multiple-choice problems, the model performance is assessed using accuracy. We label the answer options with markers like "(A)" and then determine correctness by checking the markers in the output of models. The challenge with this approach is that current LVLMs may not always adhere well to multiple-choice instructions, i.e. the output may not include the required marker.

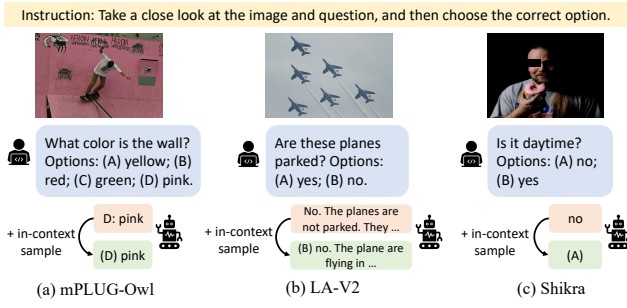

**Figure 5: Case study of the effect of in-context samples.**

To assist the evaluation of multiple-choice problems, ReForm-Eval provides both a **black-box** approach and a **white-box** approach. The black-box approach provides in-context samples to guide responses in the desired formats. Here is an example:

> $X_{\text{system-message}}$
> Human: Can you see the image? Options: (A) Yes; (B) No; (C) Not Sure; (D) Maybe.
> Assistant: The answer is (A) Yes.
> Human: $X_{\text{question}}$ Options: $X_{\text{options}}$
> Assistant: The answer is

where $X_{\text{SystemMessage}}$ is the system message required by most LVLMs, $X_{\text{question}}$ and $X_{\text{options}}$ are respectively the question and the answer options described in the text, the text in red is the in-context sample provided to the model. Notice that the in-context sample provides no information about the image. Figure 15 illustrates the effect of in-context samples with several cases. Quantitatively, with the help of the black-box strategy, the average format compliance rate of outputs from various models increases from 85% to 100%. Detailed results and analysis are provided in Table 3 and Section 6.2.

The white-box approach is based on the inherent attribute of current LVLMs as generative models. Given the visual context $v$, the question $q$, and $N$ answer options $C = \{c^i\}_{i=1}^{N}$, the answer is determined as the option with the largest generation likelihood predicted by the evaluated model:

$$\hat{c} = \arg\max_{c^i \in C} P_\theta(c^i|v, q) = \arg\max_{c^i \in C} \sum_{t=1}^{t_c} \log P_\theta(c_t^i|v, q, c_{<t}^i) \quad (1)$$

where $P_\theta(c_t^i|v, q, c_{<t}^i)$ is parameterized by the causal-LLM-based LVLMs and $\{c_1^i, ..., c_{t_c}^i\}$ is the tokenized sequence of $c^i$. For multiple-choice problems, we provide both the black-box generation evaluation results and the white-box likelihood evaluation results.

## 4.2 Instability-Aware Evaluation

As demonstrated in previous works [89, 98], LLM-based models are sensitive to different but equivalent instructions. In ReForm-Eval, instability-aware evaluation is thus introduced. For each task, multiple (more than five) instruction templates are manually designed. Each sample is tested multiple times with different templates and shuffled options if it is a multiple-choice question. The final result is based on the average of the multiple tests.

To directly characterize the instability of models, we further introduce a metric. For a multiple-choice problem with answer options $C = \{c^i\}_{i=1}^{N}$, the empirical prediction distribution of a model can be calculated from the $M$ tests as $p_i = \frac{1}{M} \sum_{j=1}^{M} \mathbb{1}(\hat{c}_j = c^i)$

| Model | Language Backbone | | Format Hit Rate | |
|---|---|---|---|---|
| | LLM Base | FT | w/o ICS | w/ ICS |
| BLIP-2$_F$ [41] | Flant5xl | No | 100 | 100 |
| mmGPT [21] | OpenFlamingo | LoRA | 95 | 100 |
| LA-V2 [19] | LLaMA-7B | Delta | 85 | 100 |
| mPLUG-Owl [91] | LLaMA-7B | LoRA | 63 | 100 |
| ImageBindLLM [24] | LLaMA-7B | No | 99 | 100 |
| InstructBLIP$_V$ [15] | Vicuna-7B | No | 100 | 100 |
| LLaVA-1.0-7B$_V$ [52] | Vicuna-7B | Full | 85 | 100 |
| Shikra [9] | Vicuna-7B | Full | 65 | 98 |
| LLaVA-1.5-7B$_V$ [50] | Vicuna-7B | Full | 96 | 100 |
| LLaVA-1.6-7B$_V$ [51] | Vicuna-7B | Full | 96 | 100 |
| ShareGPT4V-7B [10] | Vicuna-7B | Full | 94 | 100 |
| MiniGPT4 [101] | Vicuna-7B | No | 100 | 100 |
| BLIVA [26] | Vicuna-7B | No | 99 | 100 |
| PandaGPT [75] | Vicuna-7B | LoRA | 99 | 100 |
| Cheetor$_{L_2}$ [43] | LLaMA-2-7B | Delta | 99 | 100 |
| MiniGPT-v2 [8] | LLaMA-2-7B | LoRA | 100 | 100 |
| Qwen-VL-Chat [5] | Qwen-7B | Full | 95 | 100 |
| Monkey [46] | Qwen-7B | Full | 92 | 100 |
| Deepseek-VL [55] | Deepseek | Full | 88 | 100 |
| ShareGPT4V-13B [10] | Vicuna-13B | Full | 29 | 100 |
| LLaVA-1.5-13B [50] | Vicuna-13B | Full | 68 | 100 |
| LLaVA-1.6-13B [51] | Vicuna-13B | Full | 90 | 100 |
| OmniLMM-12B [1] | Zephyr-7B-β | Unk | 100 | 100 |
| Qwen-VL-Max [5] | Qwen | Unk | 100 | 100 |
| Gemini-1.0-ProV [76] | Gemini | Unk | 95 | 100 |
| GPT-4V [63] | GPT-4 | Unk | 99 | 100 |

**Table 3: The impact of in-context samples (ICS) on the format compliance rate of model outputs. Values below 95% are considered substandard. "FT" indicates whether the LLM backbone is fine-tuned. If fine-tuned, the corresponding tuning method is listed. "Unk" represents unknown.**

where $\hat{c}_j$ is the prediction of the $j$-th test. Then the instability is measured by $e = -\sum_{i=1}^{N} p_i \log(p_i)$, the entropy of the prediction distribution: Larger $e$ indicates higher uncertainty in the predictions for that sample. For text generation tasks, instability is not accessible as the prediction distribution is not directly measurable.

## 4.3 Adaptive Sub-Benchmark Construction

Many existing LVLMs utilize instruction-tuning data from task-oriented datasets, which may overlap with the data used in ReForm-Eval. To ensure fairness, we introduce two adaptive sub-benchmark construction methods: (1) **Model-oriented**: Selecting the held-out datasets common to all compared models for zero-shot evaluation; (2) **User-oriented**: Allowing users to choose and combine the benchmarks based on their own requirements. In this paper, we consider using the model-oriented method. Given several LVLMs for evaluation, we take the union of the datasets they utilize, then the sub-benchmark is ReForm-Eval excluding this union, which is the desired held-out sub-benchmark, namely ReForm-Eval-Sub. Benefiting from the broad coverage of ReForm-Eval, there still exists abundant and comprehensive evaluation data for a fair zero-shot evaluation. Unless otherwise specified, all experiments in this paper are conducted on this held-out ReForm-Eval-Sub.

| Model | Generation Evaluation | | | | | | | | | Likelihood Evaluation | | | | | | |
| | Perception | | | Cognition | | | | | R | Perception | | Cognition | | | | R |
| | CG | FG | STP | Spatial | VGR | Dialog | CMI | Desc | | CG | FG | Spatial | VGR | Dialog | CMI | |
| BLIP-2_F [41] | 69.4 | 77.4 | 36.0 | 43.2 | 73.8 | 55.5 | **51.9** | 83.6 | 11 | 60.7 | 78.4 | 51.1 | 69.3 | 53.6 | 48.8 | 13 |
| InstructBLIP_V [15] | 69.0 | 72.2 | 37.2 | 44.4 | 66.2 | 40.7 | 40.6 | 30.2 | 15 | 58.5 | 82.4 | 52.2 | 71.8 | 59.3 | 45.9 | 11 |
| LLaVA-1.0-7B_V [52] | 28.7 | 33.5 | 15.5 | 28.7 | 47.2 | 31.2 | 38.7 | 42.4 | 22 | 61.0 | 73.4 | 42.4 | 60.7 | 43.2 | 41.8 | 15 |
| MiniGPT4 [101] | 46.2 | 54.5 | 32.6 | 34.6 | 49.8 | 35.1 | 39.6 | 58.7 | 16 | 54.9 | 74.1 | 49.2 | 56.7 | 44.1 | 41.8 | 16 |
| mPLUG-Owl [91] | 41.9 | 37.8 | 36.3 | 26.8 | 40.6 | 31.9 | 37.2 | 48.4 | 19 | 57.9 | 69.1 | 48.6 | 57.8 | 38.6 | 44.1 | 17 |
| PandaGPT [75] | 28.2 | 35.4 | 1.6 | 33.3 | 49.0 | 33.4 | 37.0 | 1.1 | 25 | 42.3 | 49.9 | 39.4 | 47.0 | 39.2 | 36.6 | 23 |
| ImageBindLLM [24] | 29.2 | 33.2 | 5.3 | 35.6 | 36.9 | 33.3 | 34.6 | 32.8 | 24 | 49.6 | 56.5 | 46.1 | 52.0 | 35.9 | 39.9 | 22 |
| LA-V2 [19] | 33.2 | 30.8 | 20.5 | 23.8 | 36.4 | 32.0 | 38.0 | 41.1 | 23 | 42.7 | 64.2 | 48.6 | 58.5 | 39.5 | 43.6 | 20 |
| mmGPT [21] | 30.4 | 30.9 | 13.4 | 26.8 | 36.8 | 28.8 | 36.3 | 33.2 | 26 | 52.6 | 65.7 | 47.2 | 58.0 | 38.6 | 40.2 | 21 |
| Shikra [9] | 47.2 | 47.1 | 5.4 | 33.3 | 41.6 | 27.1 | 38.9 | 37.4 | 20 | 60.9 | 68.7 | 45.5 | 57.3 | 49.8 | 46.7 | 14 |
| Cheetor_{L_2} [43] | 46.5 | 52.9 | 17.1 | 34.5 | 59.3 | 39.8 | 39.8 | 44.4 | 17 | 52.7 | 65.0 | 48.7 | 60.7 | 41.8 | 41.0 | 18 |
| BLIVA [26] | 41.7 | 45.2 | 36.9 | 33.3 | 42.8 | 30.8 | 38.4 | 64.2 | 18 | 64.9 | 82.9 | 51.1 | 72.1 | 58.4 | 45.0 | 8 |
| LLaVA-1.5-7B_V [50] | 68.7 | 78.5 | 19.2 | 42.3 | 73.6 | 56.9 | 48.3 | 79.3 | 14 | 60.0 | 83.8 | 53.3 | 63.5 | 59.4 | 47.4 | 7 |
| MiniGPT-v2 [8] | 45.8 | 50.7 | 2.7 | 30.7 | 52.1 | 37.5 | 35.2 | 6.7 | 21 | 48.0 | 66.3 | 55.8 | 49.3 | 38.9 | 36.7 | 19 |
| Qwen-VL-Chat [5] | **73.0** | 78.5 | 38.2 | 44.6 | 73.6 | 55.6 | 49.5 | 54.6 | 10 | **69.8** | 83.5 | 50.4 | 73.1 | 61.0 | 50.7 | 4 |
| LLaVA-1.6-7B_V [51] | 69.7 | 77.0 | 21.9 | 48.3 | 75.2 | 59.8 | 48.1 | 52.6 | 13 | 61.4 | 81.9 | 53.6 | 63.8 | 58.5 | 47.5 | 11 |
| Monkey [46] | 69.0 | 75.6 | 42.0 | 45.3 | 73.2 | 48.8 | 50.4 | 54.6 | 12 | 60.8 | 81.0 | 51.3 | **74.2** | 50.2 | **50.7** | 9 |
| Deepseek-VL [55] | 68.6 | **81.4** | **43.5** | **50.0** | **75.3** | **71.2** | 49.2 | 66.3 | 6 | 56.6 | 82.4 | 54.4 | 67.1 | **63.5** | 45.1 | 10 |
| ShareGPT4V-7B [10] | 68.5 | 78.7 | 25.3 | 48.3 | 74.3 | 60.8 | 49.1 | **84.2** | 9 | 62.1 | **84.5** | 57.0 | 65.2 | 60.2 | 50.3 | 2 |
| ShareGPT4V-13B [10] | 64.3 | 81.1 | 25.7 | 55.7 | 77.5 | 67.5 | 57.0 | 91.4 | 5 | 64.0 | 83.5 | 55.9 | **71.2** | 61.3 | **50.6** | 2 |
| OmniLMM-12B [1] | **78.8** | **84.8** | **47.1** | **66.0** | **81.1** | **77.8** | **58.6** | 58.4 | 2 | 67.3 | **86.3** | **66.3** | 70.8 | **65.2** | 45.0 | 1 |
| LLaVA-1.5-13B_V [50] | 70.0 | 75.6 | 22.0 | 52.8 | 79.6 | 66.7 | 52.1 | 84.8 | 7 | 61.6 | 83.8 | 55.5 | 69.6 | 58.48 | 45.9 | 6 |
| LLaVA-1.6-13B_V [51] | 73.2 | 79.7 | 23.5 | 53.6 | 79.0 | 69.3 | 52.2 | 50.5 | 8 | 67.6 | 83.3 | 56.1 | 68.4 | 59.66 | 49.8 | 5 |
| GPT-4V [63] | 79.2 | 84.8 | 64.9 | 47.1 | 82.8 | 76.6 | 69.9 | 24.8 | 4 | - | - | - | - | - | - | - |
| Gemini-1.0-ProV [76] | 77.7 | 84.6 | 59.3 | 53.6 | 86.4 | 71.5 | 68.1 | 52.8 | 3 | - | - | - | - | - | - | - |
| Qwen-VL-Max [5] | **79.8** | **86.8** | **69.9** | **58.5** | **86.5** | **81.5** | 64.0 | **76.8** | 1 | - | - | - | - | - | - | - |

Table 4: General evaluation results of LVLMs across different capability dimensions. "CG", "FG", "CMI", and "Desc" are short for coarse-grained perception, fine-grained perception, cross-modal inference, and visual description, respectively. "R" represents the rank of average rank across capability dimensions.

## 5 EXPERIMENTS

### 5.1 Implementation Details

To demonstrate the universality of ReForm-Eval, we collect and evaluate 26 diverse LVLMs, which can be divided into open-source and proprietary API-based groups. Following LLaVA-1.6 [51], models are further divided into 3 groups: the ~7B group, the ~13B group and the proprietary group . Please refer to Appendix B.2 for a detailed introduction to these models. All experiments are conducted in the same software and hardware environment to ensure fairness. We follow the hyperparameters settings used in the original literature. Specific parameter settings are in Appendix B.1.

**Notations.** For models with multiple variants based on different backbones, we select the one with the best performance and use subscripts to denote the backbone used: $F$, $V$, $L$, and $L_2$ represent FlanT5 [14], Vicuna [13], LLaMA [78], and LLaMA2 [79], respectively. For multiple-choice problems, "Generation Evaluation" and "Likelihood Evaluation" are respectively based on the black-box and white-box strategies. Please note that likelihood evaluation is not applicable to API-based methods and text generation tasks. For each task under different strategies, the best result among each group is highlighted in **bold** while the runner-up is underlined.

### 5.2 Evaluating LVLMs with ReForm-Eval

*General Performance.* Table 4 presents the thorough performance of each model across dimensions. API-based proprietary models exhibit notable advantages, within which Qwen-VL-Max leads the pack. The overall performance of the ~13B models surpasses that of the ~7B group, implying the effectiveness of enlarging model size in multi-modal tasks. OmniLLM is the only model with comparable performance to proprietary models, but there might be concerns about potential data leakage since the training details of OmniLMM are not disclosed. Appendix C provides the complete results of performance and instability in each dimension.

*Comparison among ~7B models.* With generation-based evaluation, Deepseek-VL exhibits superior performance across multiple dimensions. Subsequently, ShareGPT4V-7B, Monkey, Qwen-VL-Chat, and LLaVA-1.6 demonstrate outstanding performance in various scenarios, with each having its strengths and weaknesses across different dimensions. In terms of likelihood evaluation, similar models stand out. Besides, the effectiveness of BLIVA, ShareGPT4V-7B, and Qwen-VL-Chat becomes apparent with likelihood evaluation, while the advantage of Deepseek-VL diminishes. We ascribe this phenomenon to the instruction-following capabilities of models. For a detailed analysis, please refer to Section 6.3.

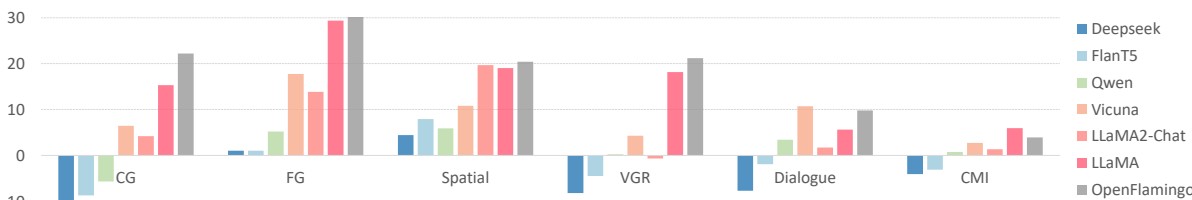

Figure 6: Comparison between two evaluation strategies. The vertical axis indicates how much the likelihood evaluation surpasses the generation evaluation. Results are grouped and averaged based on the language backbone.

Figure 7: Comparison of generation performance between different number of options in Flowers102 and Pets37.

**Comparison among ∼13B models.** OmniLMM is overwhelmingly better than the other three models, while each of those three models has its own good-performed dimensions. When evaluated under the likelihood strategy, ShareGPT-4V is slightly ahead. This phenomenon is consistent with the 7B version. Comparing the two versions of LLaVA, except for CG, the 1.6 version with expanded visual input is not significantly superior to the 1.5 version.

**Comparison among proprietary models.** Qwen-VL-Max performs the best. GPT-4V outperforms Gemini-1.0-ProV in perception tasks, yet this trend reverses in cognition tasks except for CMI.

## 6 FURTHER ANALYSIS
### 6.1 Effect of Negative Options
In this section, we illustrate the rationality of the reformulation framework of multiple-choice questions by exploring the impact of negative options. For close-vocabulary classification datasets with excessive categories, we conduct hard-negative sampling to reduce the output space. We delve into an investigation of the number of options $N$ on Flowers102 [61] and Pets37 [64]. As shown in Figure 7, with the increase of $N$, all the models' performance decreases on both datasets. $N = 4$ is a turning point, where the impact of increasing $N$ diminishes. Considering the computational cost, together with the fact that four-option multiple-choice questions are quite common, we finally set $N = 4$ for most tasks in ReForm-Eval.

In open-ended QA tasks, we explore two sources for distractors: answers from other questions in the dataset and ChatGPT. For the former source, random and text similarity-based hard-negative sampling methods are adopted. Table 5 reveals that selecting distractors from the answer pool within the dataset leads to a high false-negative rate, making them unreliable for model evaluation. Conversely, **ChatGPT-generated distractors are proved more reasonable and less prone to false negatives**, offering greater challenges compared to random options. Therefore, for open-ended QA tasks, ChatGPT emerges as the preferred source for distractors.

| Negative Options | From Dataset | | From ChatGPT |
| --- | --- | --- | --- |
| | Random | HN | |
| **FN Rate in Options** | 0.09 | 0.37 | <0.01 |
| **Avg. Score of All Models** | 45.3 | N/A | 36.9 |

Table 5: Comparison between different sources of distractors in VisDial. FN and HN are respectively short for false and hard negatives. "N/A" indicates that the corresponding experiment is omitted due to the false-negative issues.

### 6.2 Effect of In-Context Sample
To demonstrate the effectiveness of the black-box evaluation strategy introduced in Section 4.1. We assess LVLMs' ability to follow multiple-choice instructions under different strategies. The experiments are conducted in the re-formulated VQA v2, a response is considered as hitting the format if it includes the option mark like "(A)". Some results are listed in Table 3. It is obvious that the ability is tightly related to the backbone. LVLMs based on raw LLaMA inherit the weak instruction-following ability of the backbone. Additionally, fully fine-tuning the backbone entails potential risks of catastrophic forgetting of the capability, especially for 13B-based models, where the large model capacity may lead to biases towards specific response patterns. However, fine-tuning a small portion of parameters in LoRA or delta modules does not result in such issues. Nevertheless, **in-context samples effectively provide format information and guide all LVLMs to respond in the desired format**, facilitating automated evaluation.

### 6.3 Generation v.s. Likelihood Evaluation
For generation evaluation, the results reflect the coupling of the multi-modal understanding capability and the instruction-following capability. Meanwhile, likelihood evaluation directly probes the generative models and relaxes the requirement for instruction following. As shown in Figure 6, the performance gap between LVLMs under the generation and likelihood evaluation methods are tightly related to the LLM backbone. We attribute this to the capability of models to understand the multiple-choice instructions. There is a deficiency of such ability in models based on Vicuna and LLaMA, the effectiveness of these models must be demonstrated through likelihood evaluation. In contrast, models based on the other backbones, especially Deeps-eek and FLAN-T5, adapt to multiple-choice questions well. By further analyzing the relationship between options, these models can conclude with more accurate predictions. Therefore, we believe that **enhancing text comprehension ability should be emphasized in developing LVLMs**.

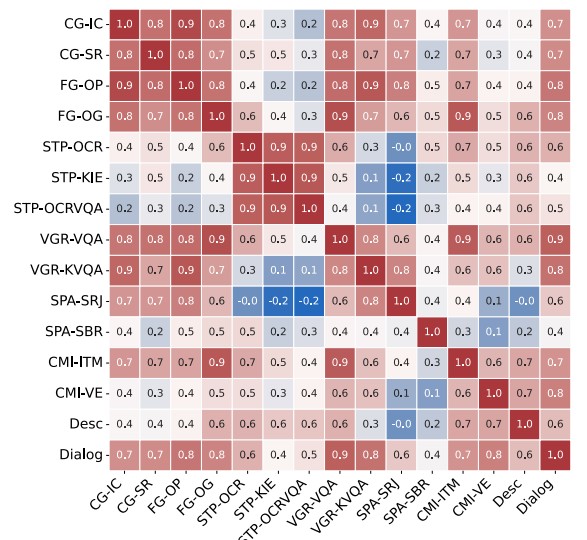

Figure 8: The correlation between different capability dimensions, calculated with performance from ~7B models. "SPA" is the short name for "Spatial" dimension.

## 6.4 Behind the Instability

To investigate the source of instability, we conduct experiments on ScienceQA by applying three types of perturbations separately to LVLMs, including random instructions, shuffling option orders, and random option marks (uppercase, lowercase, or numeric). As illustrated in Table 6, shuffling the option order results in the highest instability, highlighting a misunderstanding of the option contents. Meanwhile, we observe that most models exhibit some degree of preference for specific options (see Appendix C.6). The randomness of instruction has the least effect for each model group, suggesting that LVLMs can reasonably comprehend the carefully crafted instructions. With likelihood evaluation, the instability is overall lower because it is a white-box method that directly probes generative models without the need for random sampling during generation. **Comparison between model groups indicates that larger models are more stable.** Additionally, we reveal an apparent negative correlation between instability and accuracy in Appendix C.7. High instability to some extent reflects the model's uncertainty regarding the answer, leading to reduced accuracy.

## 6.5 Correlation among Capability Dimensions

We explore the relations among the human-crafted 8 capability dimensions by calculating their correlation coefficients. Results are shown in Figure 8. In terms of visual perception tasks, coarse-grained (CG) and fine-grained (FG) perceptions are highly correlated, whereas scene text perception (STP) operates on a largely independent axis. This suggests that **the ability to comprehend text differs significantly from the current abilities of LVLMs to process information on both local and global scales**. Moreover, the high correlations among various subtasks of STP indicate the rationality of our capability dimension design. With regard to visual cognition tasks, spatial (SPA) dimension exhibits low correlations with the other four, indicating its distinctive nature and complexity. In the interplay between visual cognition and perception tasks, SPA

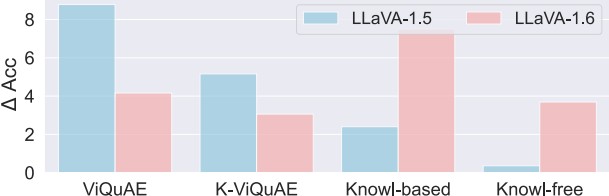

Figure 9: Performance gains from the increase of model size for Knowledge-based and Knowledge-free tasks. "Δ Acc" refers to the difference between the accuracy of the 13B model and the 7B model on the corresponding task.

| Instability Source | Generation | | | Likelihood | |
|---|---|---|---|---|---|
| | ~7B | ~13B | Pro. | ~7B | ~13B |
| **Instruction** | 0.13 | **0.05** | 0.12 | **0.06** | **0.06** |
| **Option Order** | 0.51 | 0.26 | **0.22** | N/A | N/A |
| **Option Mark** | 0.23 | **0.08** | 0.12 | N/A | N/A |

Table 6: Average instability by three types of random perturbations across distinct groups. The calculation formula is defined in Section 4.2. "Pro." represents the proprietary group.

demonstrates a stronger connection with FG than CG. This suggests that **spatial abilities are tailored to fine-grained visual details**. Furthermore, the correlation between STP and SPA is relatively low, and both dimensions also show little correlation with other capability dimensions. In the future, models can improve their ability in spatial analysis and scene text perception through the relevant datasets provided by our ReForm-Eval benchmark.

## 6.6 Knowledge in Model Capacity

The richness of ReForm-Eval empowers comparative analysis from a specific perspective. We take the exploration of the LVLMs' internal knowledge as an example. We select two subsets: tasks requiring additional knowledge and knowledge-free tasks, along with a strictly controlled group of ViQuAE and its variant K-ViQuAE (where the knowledge is provided in the question). Figure 9 illustrates the performance gain across different tasks with the increase in model capacity. The improvement is more pronounced for knowledge-based tasks, **verifying the effectiveness of scaling up model sizes in expanding internal knowledge of LVLMs.**

## 7 CONCLUSION

In this paper, we propose to re-formulate task-oriented multi-modal benchmarks to evaluate LVLMs. By efficiently re-formulating 61 benchmarks into unified formats, we construct a benchmark, namely ReForm-Eval, covering 8 capability dimensions. Compared with previous benchmarks for LVLMs, ReForm-Eval provides more data without the need for manual annotation. We further design a dependable automated evaluation framework, ensuring an impartial assessment of different LVLMs. Leveraging ReForm-Eval, we conduct exhaustive evaluations of various LVLMs and delve into the factors influencing their performance. Generally speaking, we believe ReForm-Eval serves as a reliable tool for quantitative analysis of LVLMs, aiding in the research and development of LVLMs.

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
