# OpenReview forum: "ReForm-Eval: Evaluating Large Vision Language Models via Unified Re-Formulation of Task-Oriented Benchmarks"
_acmmm.org/ACMMM/2024/Conference — MM2024 Poster_

### Official Review · Reviewer_Yuod · 2024-05-20

**Rating:** 3
**Confidence:** 3

**Summary:**

This paper proposes a unified framework of evaluation on LVLMs, aiming to bridge the gap between the structured output of task-oriented benchmarks and the free-form output by the nature of LVLMs. Multiple benchmarks are mixed and universally evaluated with ReForm-Eval on 26 LVLMs to show the effectiveness.

**Strengths:**

1. The motivation on bridging the gap between structured output from task-oriented benchmarks and free-form output from LVLMs seems makes sense. It is good if there is a unified framework to 1) not limit the form of the outputs from LVLMs; 2) yet can precisely semantically capture the answers from LVLMs despite the free forms; 3) can be plugged into arbitrary task-oriented benchmarks, i.e., plug-and-play.

2. The presentation is clear and easy-to-read.

3. The examples in Fig. 1 are desired and match the contribution claim.

**Limitations:**

[The motivation is a fancy dream while the method is too naive.] The authors seem just manually classify the current tasks into two output formats: multiple-choice and text generation. And for each, the authors still conduct conventional structuring methods such as in-context learning to enforce LVLMs outputting structured output. In the other words, the only difference from previous works is that the authors mix multiple benchmarks together, arange them with different output structures, and proceed with conventional evaluation formats. I think it is much more interesting if the authors can devise a method that, given the free-form answers from LVLMs and structured labels from benchmarks, ReForm-Eval may semantically determine it is a correct answer or not, instead of changing the outputs of LVLMs depending on what the task is. The current solution is no different compared to just averaging results across multiple task-oriented benchmarks.

[The experiments do not exhibit the advantage of ReForm-Eval.] Although the authors conduct substantial experiments evaluating more than 20 LVLMs with ReForm-Eval, I do not see any advantages of ReForm-Eval compared to other benchmarks. And the remaining studies such as the effect of in-context examples may be well-studied in some works such as MMICL[1], and not the main advantage of a benchmark paper.

[Some points to consider.] Depending on the interesting motivation proposed by the authors, i.e., bridging the gap between the structured outputs and free-form outputs, I wonder if the structured outputs of LVLMs enforced by methods such as in-context learning compromise the accuracy of LVLMs in current task-oriented benchmarks. In the other words, if the authors may devise a new method to detect the free-form texts reflect the correct answers or not, and outperforms the current conventional structured outputs, there may be a performance gap between the two and the enforced structured outputs of LVLMs may compromise their performance. For instance, enforcing LVLMs outputting the option "(A) yes" instead of some free-form texts such as "Yes, it is" may be harmful? This is an open-ended question and it is interesting to know.

[Summary] This is an acceptable paper if the authors can add more contributions to the motivation and answers the questions as mentioned above. Overall it is promising yet not ready as an impactful work.

[1] MMICL: Empowering Vision-language Model with Multi-Modal In-Context Learning, ICLR2024

**Suitability:**

2

---

### Official Review · Reviewer_hBS3 · 2024-05-24

**Rating:** 6
**Confidence:** 4

**Summary:**

This paper introduces ReForm-Eval, a unified benchmark for comprehensively evaluating Large Vision Language Models (LVLMs). ReForm-Eval re-formulates 61 existing task-oriented datasets into multiple-choice questions or text generation problems that are compatible with the free-form output of LVLMs. The benchmark covers a wide range of capabilities including visual perception, spatial understanding, visually grounded reasoning, cross-modal inference, and multi-turn dialogue. The authors also propose an evaluation framework that considers the input sensitivity and output control of LVLMs. Extensive experiments are conducted to assess various state-of-the-art LVLMs using ReForm-Eval.

**Strengths:**

1. ReForm-Eval efficiently re-formulates existing datasets into LVLM-compatible formats, enabling the utilization of abundant existing resources for evaluation.

2. The benchmark covers a wide range of capability dimensions and tasks, providing a comprehensive assessment of LVLMs.

3. ReForm-Eval offers a large-scale evaluation resource with over 500,000 samples, surpassing previous benchmarks that rely on manual annotation.

4. The proposed evaluation framework is designed to be reliable and fair, considering factors such as input sensitivity and output control of LVLMs.

5. The paper conducts extensive experiments and analysis using ReForm-Eval, providing valuable insights into the performance and characteristics of various LVLMs.

**Limitations:**

1. The re-formulation process may introduce noise or bias compared to the original datasets, potentially affecting the evaluation results.
2. The benchmark primarily focuses on vision-language tasks and may not cover all possible multimodal scenarios or capabilities of LVLMs.
3. The evaluation metrics used in the paper, such as accuracy and instability, may not fully capture the nuances and qualitative aspects of LVLM performance.
4. The paper does not provide detailed ablation studies or analysis on the impact of different re-formulation strategies or evaluation design choices.
5. The benchmark and evaluation framework may require updates and extensions in the future to keep pace with the rapid development of LVLMs and emerging task requirements.
6. MMT-Bench[1] is a new multitask benchmark which also repurposes a lot of open-source benchmarks to evaluate the LVLM in a multimodal multitask manner. The author should mention this in the submission. In addition, it is necessary to mention [2] and [3] in the paper.

[1] Ying, Kaining, et al. "MMT-Bench: A Comprehensive Multimodal Benchmark for Evaluating Large Vision-Language Models Towards Multitask AGI." arXiv preprint arXiv:2404.16006 (2024).

[2] Shao, Wenqi, et al. "Tiny lvlm-ehub: Early multimodal experiments with bard." arXiv preprint arXiv:2308.03729 (2023).

[3] Liu, Shuo, et al. "ConvBench: A Multi-Turn Conversation Evaluation Benchmark with Hierarchical Capability for Large Vision-Language Models." arXiv preprint arXiv:2403.20194 (2024).

**Suitability:**

3

---

### Official Review · Reviewer_jQTs · 2024-05-25

**Rating:** 3
**Confidence:** 3

**Summary:**

This paper introduces a novel benchmark called ReForm-Eval for evaluating Large Vision Language Models (LVLMs). The authors present a systematic data collection and reformulation process to convert 61 existing benchmark datasets into unified formats compatible with LVLMs, spanning various capability dimensions from basic visual perception to advanced visual reasoning and dialogue. ReForm-Eval aims to make full use of public resources to provide a large-scale, easy-to-use, universal, and efficient evaluation approach. Additionally, the paper demonstrates the comprehensiveness and reliability of ReForm-Eval through extensive experiments and analyses across various LVLMs.

**Strengths:**

1. This paper introduces an innovative benchmark, ReForm-Eval, which unifies the evaluation of LVLMs by re-structuring existing benchmarks.
2. This paper presents an automated strategy that reduces the need for manual annotation and offers a fair assessment across models.

**Limitations:**

1. There is too much data in this benchmark, which will prolong the evaluation process. In addition, the original intention of bmk designation should be to cover as much data distribution as possible on the limited data.
2. Whether the evaluation framework can fully reflect the generalization ability of LVLMs in practical applications remains to be verified, and the indicators and somatosensory are not aligned.

**Suitability:**

2

---

### Meta-Review · Area_Chair_8saX · 2024-07-05

**Recommendation:** Accept (Poster)
**Confidence:** 5

**Metareview:**

There are divergent ratings after rebuttal. jQTs did not give its final recommendation, but according to the authors' response, the concerns of the data scale and the evaluation of zero-shot capability of the benchmark should have been addressed. Yuod concerned that the reformulation process may require further study, such as the gap between the proposed output formats and the motivations behind them. AC agrees that the proposed reformulation process can be improved, but current contributions have met the bar of ACM MM. Therefore, after considering all reviews, the rebuttal, and the subsequent discussion, the consensus is to accept the paper.